# Semi-automated closed system manufacturing of lentivirus gene-modified haematopoietic stem cells for gene therapy

Jennifer E. Adair[1,2], Timothy Waters[3], Kevin G. Haworth[1], Sara P. Kubek[1], Grant D. Trobridge[4], Jonah D. Hocum[4], Shelly Heimfeld[1] & Hans-Peter Kiem[1,2]

Haematopoietic stem cell (HSC) gene therapy has demonstrated potential to treat many diseases. However, current state of the art requires sophisticated *ex vivo* gene transfer in a dedicated Good Manufacturing Practices facility, limiting availability. An automated process would improve the availability and standardized manufacture of HSC gene therapy. Here, we develop a novel program for semi-automated cell isolation and culture equipment to permit complete benchtop generation of gene-modified CD34$^+$ blood cell products for transplantation. These cell products meet current manufacturing quality standards for both mobilized leukapheresis and bone marrow, and reconstitute human haematopoiesis in immunocompromised mice. Importantly, nonhuman primate autologous gene-modified CD34$^+$ cell products are capable of stable, polyclonal multilineage reconstitution with follow-up of more than 1 year. These data demonstrate proof of concept for point-of-care delivery of HSC gene therapy. Given the many target diseases for gene therapy, there is enormous potential for this approach to treat patients on a global scale.

[1] Clinical Research Division, Fred Hutchinson Cancer Research Center, 1100 Fairview Avenue North, Seattle, Washington 98109, USA. [2] Departments of Medical Oncology and Pathology, University of Washington, 1410 Campus Parkway, Seattle, Washington 98195, USA. [3] Miltenyi Biotec Inc., 2303 Lindbergh St, Auburn, California 95602, USA. [4] Department of Pharmaceutical Sciences, Washington State University Spokane, PO Box 1495, Spokane, Washington 99210, USA. Correspondence and requests for materials should be addressed to J.E.A. (email: jadair@fhcrc.org).

There is tremendous potential for haematopoietic stem cell (HSC) and progenitor (CD34+) cell gene therapy for many diseases (reviewed in refs 1,2), but as the field closes in on large global health burdens such as HIV and haemoglobinopathies, lack of a portable technology for standardized manufacture of gene-modified CD34+ blood cell products becomes a critical barrier to widespread clinical use. Certainly in vivo genetic modification would make this treatment highly portable, and preclinical studies are currently underway[3–7]. However, this approach has some disadvantages: (1) for many disease targets, conditioning is required to provide an engraftment advantage to gene-modified cells; (2) there is unknown risk associated with genetic modification of off-target cell types; and (3) there is limited ability to achieve therapeutic levels of genetic modification in the target CD34+ cell population (reviewed in ref. 8). Ex vivo lentivirus vector (LV)-mediated gene transfer into CD34+ haematopoietic cells is the most clinically successful method applied to date, permitting subsequent development of all blood cell types for the lifetime of the patient. Recently, more targeted gene editing approaches are being developed to ameliorate-risks associated with semi-random retrovirus genomic insertion (reviewed in ref. 2). However, regardless of the method of genetic modification, ex vivo manipulation of CD34+ haematopoietic cells introduces the risk of contamination with infectious agents and reduces engraftment potential and haematopoietic fitness[9–12]. Thus, a short ex vivo manipulation protocol in a closed system would represent a significant advance in the field, permitting distribution beyond a small number of sophisticated centres.

Ex vivo manufacturing generally includes (1) immunomagnetic bead-based isolation of target CD34+ cells, (2) CD34+ cell supportive culture conditions with (3) defined gene modification reagents and conditions and finally, (4) removal of residual manufacturing reagents for preparation and testing of the final cellular product for infusion. All of these steps are carried out under current Good Manufacturing Practices (cGMP), but the CD34+ cell source (that is, bone marrow (BM) or growth factor mobilized leukapheresis (HPC-A)), and the therapeutic genetic modification vary depending on the target patient population. Here we sought to develop a closed system, automated manufacturing platform with minimal user interface, which could accomplish all of the steps in the ex vivo manufacture of genetically modified CD34+ cells from start to finish, while meeting cGMP criteria.

We previously demonstrated efficient ex vivo CD34+ cell LV-mediated gene transfer in less than 36 h as part of a gene therapy program for Fanconi anaemia (FA)[13]. FA CD34+ cells are rare and respond poorly to mobilization[14]. Thus a phase I trial utilizing BM as the CD34+ cell source was initiated (National Clinical Trials registry ID: NCT01331018). However, FA BM products require removal of unwanted red blood cells (RBC) by gentle sedimentation in hetastarch (HES)-based media without centrifugation[15].

To accomplish this, an HES sedimentation protocol for up to 1.8 l of BM was developed using customized programming for the CliniMACS Prodigy device (Miltenyi Biotec GmbH). This commercially available device permits automated pre-processing, immunomagnetic labelling and separation of target cells, including CD34+ cells and T cells, from human HPC-A products[16,17], and is capable of large scale, automated Ficoll-based RBC depletion from BM[18].

It was then hypothesized that a point-of-care strategy for patient-specific CD34+ cell gene transfer could be designed on this device, eliminating the need for local cGMP facility infrastructure. The overall goal for proof-of-concept was rapid, mostly automated production of ex vivo LV gene-modified

patient-specific CD34+ cell products suitable for human infusion and haematopoietic repopulation.

Here we demonstrate that this semi-automated benchtop system can enrich and transduce CD34+ cells from both BM and HPC-A products with minimal user input. The yield, purity and rates of transduction of the CD34+ cells are comparable to current cGMP practices, and pass cGMP standards for human-transduced products. These transduced cell products are capable of engrafting in both immunodeficient mice in a xenograft model, as well as reconstituting polyclonal, multilineage haematopoiesis in a myeloablative nonhuman primate (NHP) transplant model. These data demonstrate the potential to provide cell products for gene therapy to patients unreachable by the current state-of-the-art GMP facilities.

## Results
All experiments were conducted in a non-cGMP laboratory with a benchtop device. Additional equipment included a biosafety cabinet, sterile tubing welder, general laboratory equipment and personal protective equipment. Our first challenge was to reconfigure this technology to complete all of the manufacturing steps following CD34+ cell enrichment. Initial programming was scripted using extensible markup language as separate functions including product preparation and labelling, immunomagnetic cell separation, culture, fluid addition, spinoculation, sampling, harvest and formulation. Since the user defines whether these programs are used and in what order, and must intermittently interface with the device during program launch and certain inter-program processes, we use the term 'semi-automated' rather than 'fully automated' to describe this method of manufacture. A complete list of programs developed and the component processes encoded in each is listed in Supplementary Table 1 (programs available upon request).

**Manufacture of LV-transduced CD34+ cells from leukapheresis.** To test these novel programs, granulocyte colony stimulating factor (G-CSF) mobilized HPC-A products collected from healthy human (Hu) donors (HPC-A donor 1, HPC-A donor 2 and HPC-A donor 3) were processed first, since HPC-A is the most common source of CD34+ cells in adult gene therapy patients. Automated manufacturing included platelet washes before CD34+ cell labelling (Supplementary Table 1). Complete manufacturing required four custom program modules, with execution lasting ∼25 h from receipt of the starting HPC-A product to completion (that is, release) of the final product for infusion (Table 1). Two different, hand-fabricated tubing sets were needed to access all required device components for processing (see Methods and Supplementary Table 1). A mean 62.3% yield of CD34+ cells from HPC-A products with purities ≥96% was achieved (Supplementary Table 2). Absolute yields were $1.4 \times 10^8$, $3.6 \times 10^8$ and $2.3 \times 10^8$ CD34+ cells for each product, respectively. Because of chronology of HPC-A product receipt and limited clinical grade vector available for transduction, LV-mediated gene transfer was only performed in the first two products (Hu HPC-A donor 1 and Hu HPC-A donor 2).

The gene transfer vector tested was an anti-HIV LV vector currently being tested in a phase I clinical trial at our institution for patients with AIDS-related lymphoma [NCT02343666]. It is a commonly used self-inactivating (SIN) HIV-1-derived backbone encoding three therapeutic transgenes: C46, a 46-amino acid peptide derived from gp41, which inhibits the HIV membrane fusion; a short hairpin targeting the chemokine receptor 5 (CCR5) for degradation via cellular RNA interference pathways; and a P140K mutant methylguanine methyltransferase (MGMT) trans-gene[19] to allow for in vivo selection of gene-modified cells if poor

**Table 1 | Total time and hands-on operator time required for each semi-automated process.**

| Source product | Start time (day −1) | End time (day 0) | Total process time (h) | Mean total process time (h) | Total hands-on time required (h:min) | Mean hands-on time required (h:min) |
|---|---|---|---|---|---|---|
| Hu HPC-A (donor 1) | 11:19 | 11:50 | 25 | 25 | 2:10 | 3:25 |
| Hu HPC-A (donor 2) | 11:10 | 11:50 | 25 | 25 | 4:39 | 3:25 |
| Hu BM (donor 1) | 07:15 | 10:04 | 27 | 27.5 | 2:56 | 2:59 |
| Hu BM (donor 2) | 08:00 | 13:38 | 28 | 27.5 | 3:01 | 2:59 |
| NHP BM (Z13105) | 07:58 | 15:50 | 32 | 30 | 4:45 | 3:44 |
| NHP BM (Z13083) | 09:30 | 13:26 | 28 | 30 | 2:43 | 3:44 |

BM, bone marrow; NHP; nonhuman primate.

engraftment is observed (Supplementary Fig. 1a). The multiplicity of infection (MOI) was 20 infectious units (IU) per cell and vector exposure was 12–14 h with no pre-culture to stimulate cell division (see Methods). In addition, the pyrimidoindole derivative, UM729, was included during transduction as this molecule is reported to expand primitive haematopoietic cells *ex vivo*[20].

Donor-to-donor variation in mean vector copy number (VCN) per cell was observed in liquid cultures maintained for 10 days (2.1 and 10.1 vector copies cell$^{-1}$, respectively) (Supplementary Fig. 2). The high VCN observed for HPC-A donor 2 is undesirable for clinical purposes due to risk of insertional mutagenesis[21]. Efficiency of gene transfer into colony-forming cells (CFC) was 77.7 and 59.3%, respectively (Supplementary Fig. 2a,b). To assess product fitness, release testing and xenotransplantation of resulting cell products into non-obese diabetic/severe combined immunodeficiency gamma$^{-/-}$ (NSG) mice following sub-lethal total body irradiation (TBI) were conducted (Fig. 1a). Both products met cGMP immediate release criteria for infusion (Supplementary Table 3). Robust engraftment of human CD45$^+$ cells from both products was observed in peripheral blood (PB) and BM of xenotransplanted mice over 12 weeks following infusion (Fig. 1b). At 12 weeks, mice were killed and human cell engraftment in BM was determined (Fig. 1c). Mean VCN per human blood cell in mice was 0.083 and 1.7 for HPC-A donor 1 and HPC-A donor 2, respectively (Fig. 1d). Mean VCN per cell was higher in mice receiving HPC-A donor 2 cells, consistent with the higher *in vitro* VCN value. These data indicate isolation and overnight LV transduction of repopulating CD34$^+$ cells from mobilized HPC-A products is possible with this automated, closed system platform.

To assess whether UM729 contributed to high VCN values, we repeated transduction of CD34$^+$ cells from the same two HPC-A donors without UM729. We observed mean VCN of 1.8 and 5.0 vector copies per cell after 10-day culture, respectively for HPC-A donor 1 and HPC-A donor 2 (Fig. 2).

**Manufacturing of LV-transduced CD34$^+$ cells from donor BM.** The semi-automated process was then expanded for BM CD34$^+$ cell products, which are used in gene therapy for very young patients and/or disease settings where mobilization is contra-indicated, such as FA and sickle cell disease. Three healthy adult BM donors were collected and programming was modified for CD34$^+$ cell selection following HES sedimentation of RBC. This custom programming achieved >91% depletion of RBC and ≥57% yield of CD34$^+$ cells in BM, with subsequent automated labelling of CD34$^+$ cells in <6 h in a closed system. (Supplementary Table 4). This lengthened total manufacturing time, but maintained reduced hands-on operator time required (Table 1). An average 49.5% yield of the starting CD34$^+$ cell population at >72% purity was achieved following HES sedimentation and CD34 enrichment (Supplementary Table 5).

The first two products (Hu BM donor 1 and Hu BM donor 2) were transduced.

The same anti-HIV LV vector was used for transduction of BM CD34$^+$ cells. For Hu BM Donor 1 a total of $8 \times 10^6$ CD34$^+$ cells were obtained from 96 ml of BM, resulting in sub-optimal cell density ($2.0 \times 10^5$ cells per ml). In addition, low gas exchange during the culture period, visualized as dark pink media coloration, was observed. Here, the total cell number expanded during culture, thus we observed an increase in the absolute number of viable cells (from $8 \times 10^6$ to $12 \times 10^6$ cells). However, the overall viability of the resulting cell product as measured by trypan blue dye exclusion declined from 86% at initial culture to 40% following transduction. Mean VCN in 10-day liquid cultures was 10.6 copies per cell (Supplementary Fig. 2c), and CFC content was 1.5% for this product, with 36.4% transduced CFCs (Supplementary Fig. 2d).

The process was repeated with Hu BM donor 2, with $21.8 \times 10^6$ CD34$^+$ cells obtained, permitting optimal cell density during transduction ($0.5–1 \times 10^6$ cells per ml). VCN in 10-day cultures was 0.3 vector copies per cell (Supplementary Fig. 2c), with CFC content and transduction efficiency (3.0 and 43%, respectively) modestly increased compared with Hu BM Donor 1 (Supplementary Fig. 2d). All release test criteria were met (Supplementary Table 4), and xenotransplantation into NSG mice resulted in stable engraftment of human CD45$^+$ blood cells up to 14 weeks after infusion (Fig. 3). BM engraftment of human CD45$^+$ and CD34$^+$ white blood cells ranged from 5 to 15% (Fig. 3b). Up to 16% of BM cells were LV gene-modified in these animals (Fig. 3c). These data demonstrate successful manufacturing of gene-modified CD34$^+$ cells from BM products using the semi-automated platform.

**Autologous reconstitution of myeloablated nonhuman primates.** Given the potential for increased exposure to infectious agents in a non-cGMP laboratory setting, we wanted to rigorously evaluate the safety and fitness of autologous cell products produced in the semi-automated system in a myeloablative transplantation setting. We chose the pigtailed macaque (*Macaca nemestrina*) transplantation model (reviewed in ref. 22). This model uses the same growth factors and media used in human cell product manufacture. However, directly conjugated clinical reagents for CD34$^+$ immunomagnetic selection are not cross-reactive with macaque cells. Therefore a modified two-step program was created for indirect bead labelling by mouse anti-human primary antibody labelling and then secondary rat anti-mouse immunoglobulin M (IgM)-conjugated microbead affinity.

We manufactured autologous LV gene-modified NHP CD34$^+$ cell products using the semi-automated platform and trans-planted these into two animals (Z13105 and Z13083) following myeloablative TBI (1020 cGy). Total manufacturing time required for each NHP BM product averaged 30 h with <4 h of direct operator hands-on time (Table 1). Two-step immunomagnetic

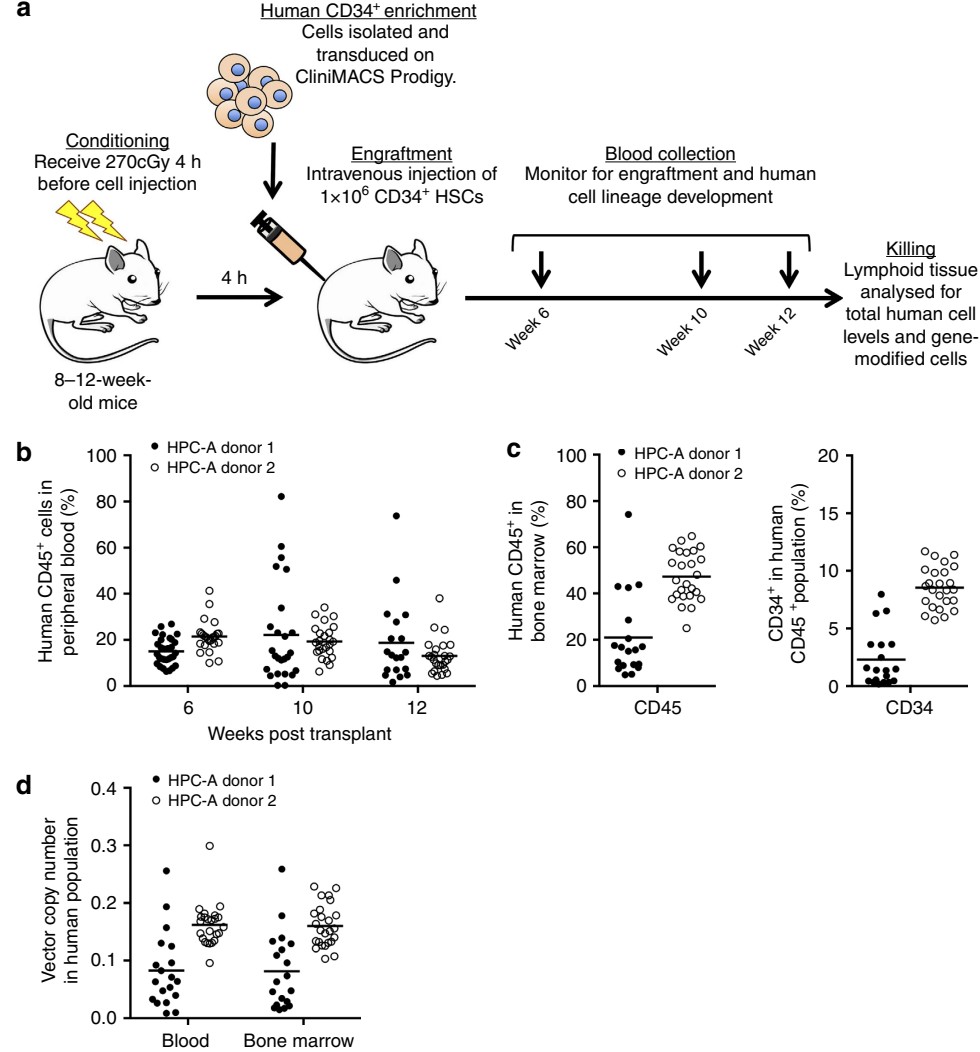

**Figure 1 | Human mobilized gene-modified CD34$^+$ cells produced using point-of-care manufacturing engraft in immunodeficient mice.** (**a**) Human CD34$^+$ cells were obtained from mobilized PB after HPC-A. Adult NSG mice ranging between 8 and 12 weeks of age received a sub-lethal dose (270 cGy) of radiation 4 h before intravenous injection of $1 \times 10^6$ gene-modified human CD34$^+$ cells. Mice were then followed for 12 weeks post transplant. At 12 weeks, animals were killed and lymphoid tissues were analysed for total human cell levels as well as frequency of genet modification. (**b**) Human CD45$^+$ cell engraftment levels in PB of individual adult NSG mice receiving LV-transduced CD34$^+$ cells from human apheresis (HPC-A) donors. (**c**) At 12 weeks after injection, BM was analysed for both total human CD45$^+$ cell content and human CD34$^+$ levels. (**d**) The level of gene-modified human cells in both PB and BM of each recipient was determined by quantitative PCR.

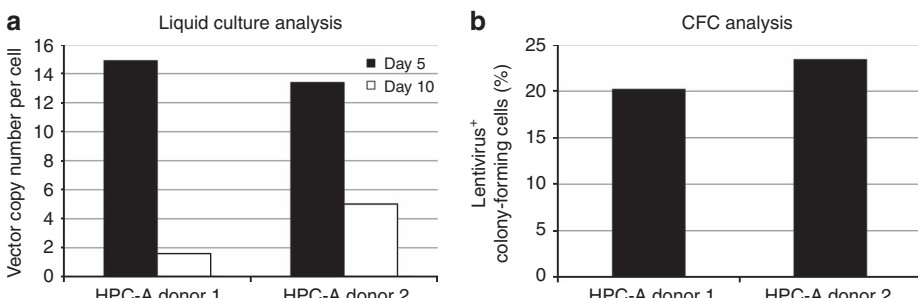

**Figure 2 | Removal of UM729 from transduction conditions reduces VCN but also CFC transduction efficiency.** Following non-automated transduction of CD34$^+$ cell products from HPC-A donors 1 and 2 in the absence of UM729, aliquots of the final cell product were cultured in liquid media consisting of recombinant human growth factors G-CSF, SCF, TPO, Flt3-L, IL-3 and IL-6 for subsequent real-time PCR to determine VCN (**a**), or in methylcellulose media containing the same recombinant human growth factors for colony-forming assay (**b**).

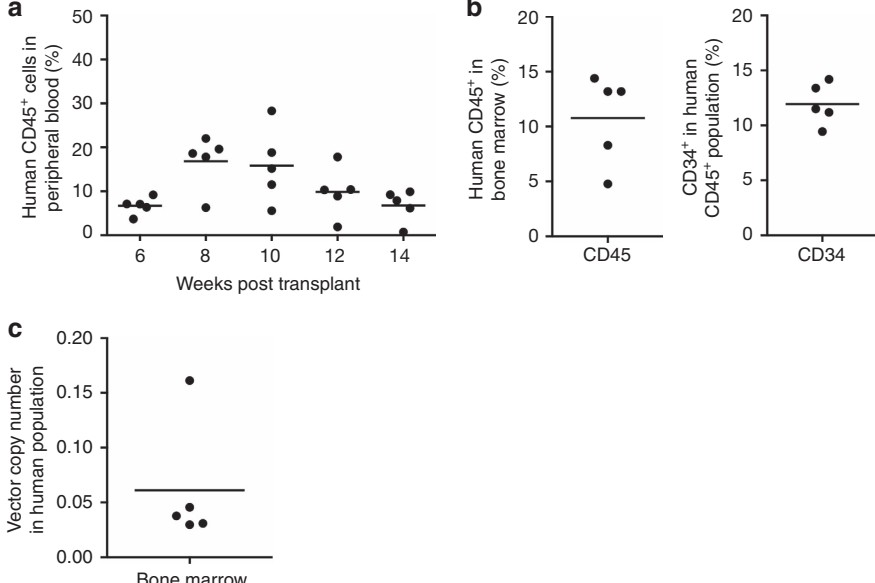

**Figure 3 | Human steady-state BM CD34$^+$ cells produced using point-of-care manufacturing engraft into immunodeficient mice. (a)** Human CD45$^+$ blood cell engraftment levels at various time points post injection in individual adult NSG mice receiving LV-transduced BM-derived CD34$^+$ cells. All animals were killed at 14 weeks following transplant and BM was analysed for human CD45$^+$ and CD34$^+$ blood cell content (**b**), as well as LV gene marking (**c**).

bead labelling resulted in lower CD34$^+$ cell yields from NHP BM products (Supplementary Table 6); however, cell yields were sufficient for transplantation of juvenile monkeys weighing <4 kg. The LV vector used here was the same HIV-1-derived vector backbone, but encoding an enhanced green fluorescent protein (*eGFP*) transgene to facilitate *in vivo* tracking, and a P140K mutant *MGMT* transgene (Supplementary Fig. 1b). A MOI of 40 IU per cell was used during a two-hit transduction protocol (20 IU per cell × 2). Total cell doses achieved were $27 \times 10^6$ and $5.4 \times 10^6$ CD34$^+$ cells per kg body weight, respectively, reflecting a gain in the absolute number of cells after transduction in animal Z13105 (from $7.2 \times 10^7$ to $8.2 \times 10^7$ total cells), and a loss for animal Z13083 (from $3.0 \times 10^7$ to $1.9 \times 10^7$ total cells). Transduction efficiency in CFC was 23 and 39%, respectively, with 27.6 and 11.9% of cells in 11-day cultures expressing *eGFP* (Supplementary Fig. 3). Engraftment, defined as sustained absolute neutrophil count (ANC) >500 mm$^{-3}$ and platelet count >20,000 mm$^{-3}$, occurred within +23 days from transplant for both animals (Fig. 4), which is comparable to engraftment observed in animals that were infused with autologous CD34$^+$ haematopoietic cells transduced with the same LV vector under standard, operator-dependent conditions (Table 2). Notably, neither animal required unanticipated supportive care or displayed increased toxicity, including potential contamination, following benchtop production of genetically modified infused cell products.

Stable, persistent gene marking in PB granulocytes and lymphocytes was documented by eGFP protein expression within 1 month after infusion in both animals, persisting for more than 1 year and reaching levels of ~12% (Fig. 4a), consistent with gene-modified cell levels in animals transplanted with LV-transduced cells manufactured by a standard, operator-dependent protocol on fibronectin-coated flasks (Table 2). At 6 months after transplant, we observed up to 1.7% GFP$^+$ RBC and 1.4% GFP$^+$ platelets in these two animals (Supplementary Fig. 4). As successful anti-HIV gene therapy requires gene-modified T cells in particular, we tracked lymphocyte reconstitution after transplant (Fig. 3b). Early after transplant (<100 days), nearly

all eGFP$^+$ lymphocytes were CD20$^+$ B cells in both animals with increasing contributions of eGFP-expressing CD3$^+$, CD4$^+$ and CD8$^+$ lymphocytes observed at >100 days. An unexpected reduction in eGFP$^+$ granulocytes (animal Z13105) beginning at +76 days after transplant, corresponded to below-target tacrolimus dosing to prevent immune response against eGFP (Fig. 4a). Dose re-targeting rebounded eGFP$^+$ granulocytes to 27%. Clonal diversity of eGFP$^+$ cells demonstrated highly polyclonal distribution of gene-modified PB leukocytes in each animal without the need for chemotherapy-induced selection of gene-modified cells *in vivo* (Fig. 4c). This demonstrates safety and feasibility of point-of-care manufacturing (Supplementary Fig. 5) of LV gene-modified CD34 + cells in a clinically relevant large animal model.

## Discussion

Here we demonstrate overnight manufacturing of LV gene-modified CD34$^+$ haematopoietic repopulating cells in a portable, self-contained benchtop platform suitable for point-of-care implementation. These manufactured human BM and mobilized HPC-A products meet current regulatory requirements for infusion in gene therapy clinical trials and are capable of *in vivo* reconstitution in an immunodeficient mouse xenotrans-plantation model. Importantly, two myeloablated monkeys were successfully reconstituted for an extended period of time with autologous LV gene-modified CD34$^+$ cells produced using this system, without any adverse events related to the benchtop manufacturing methods used.

The range of blood stem cell gene therapy applications is expanding rapidly for inherited diseases, HIV/AIDS and cancer (reviewed in refs 1,23). Most studies are conducted at single institutions, limited to roughly one dozen manufacturing facilities in seven countries (USA, Spain, France, UK, Italy, Germany and Australia). Thus distribution is a major barrier to translation of gene therapy into clinical practice. The flexibility and small footprint of this technology provides a major advantage to scale up of the current state-of-the-art cell manufacturing. Rather than

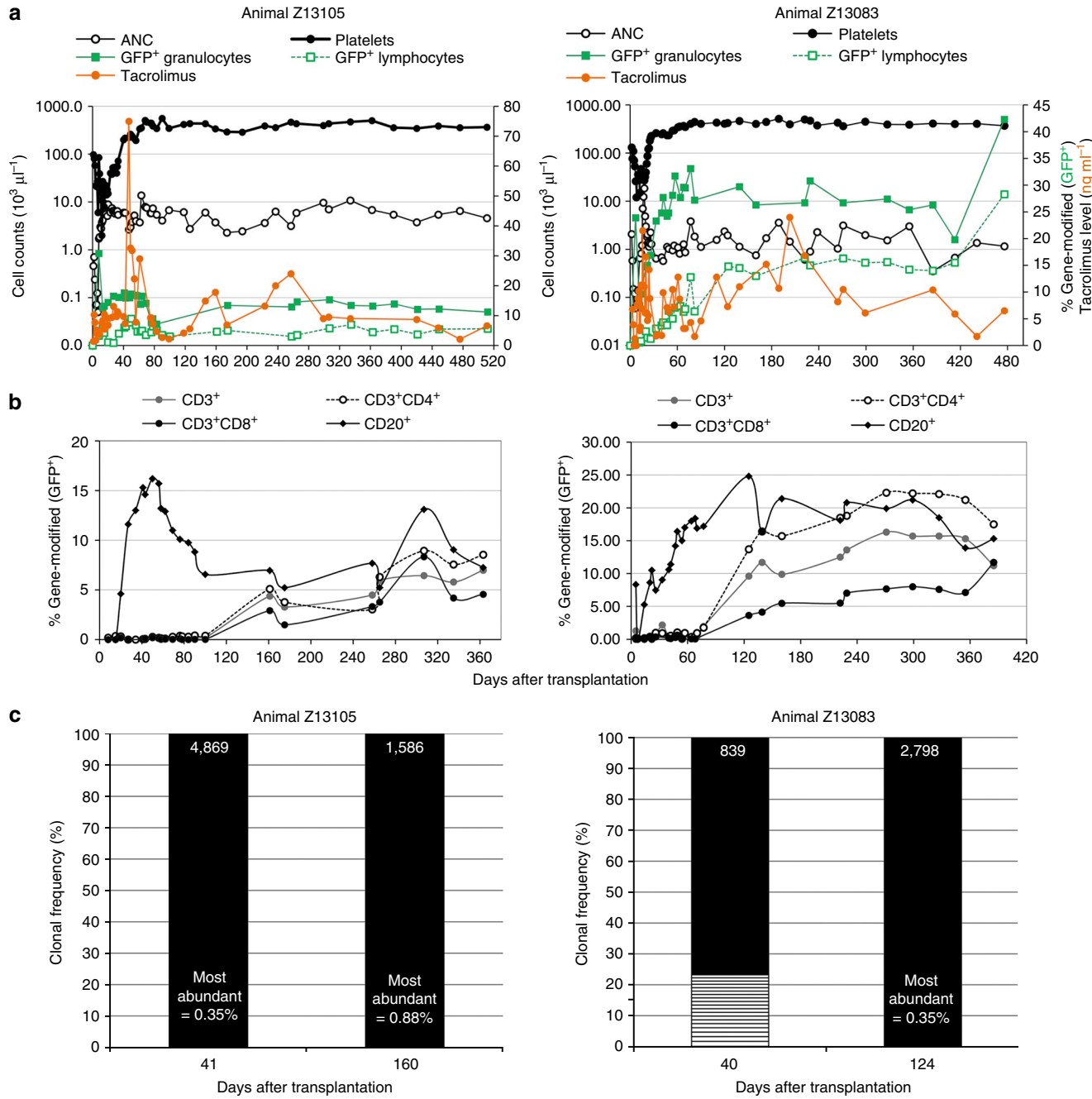

**Figure 4 | Sustainable haematopoiesis and engraftment of LV gene-modified CD34$^+$ cells *in vivo* in NHPs following point-of-care manufacturing.** Two animals (Z13105 and Z13083) received autologous, LV gene-modified CD34$^+$ cells produced under semi-automated conditions following myeloablative TBI. (**a**) Graphs depict haematopoietic recovery by ANCs (open black circles) and platelet counts (closed black circles) on the primary *y* axis, engraftment of gene-modified PB granulocytes (closed green squares) and lymphocytes (open green squares) and measured tacrolimus levels in serum (closed orange circles) on the secondary *y* axis as a function of time after transplantation (*x* axis). (**b**) Percent of gene-modified (eGFP$^+$) lymphocytes expressing CD3, and the percentage of eGFP$^+$/CD3$^+$ cells expressing CD4 and/or CD8 observed in PB (*y* axis) over time after transplant (*x* axis). (**c**) Highly polyclonal engraftment of LV gene-modified NHP CD34$^+$ haematopoietic cells following point-of-care production and transplant. Bar graphs represent the clonal diversity of LV gene-modified PB leukocytes collected at day +40 after transplant into autologous recipients as determined by genomic locus of integration. Clonal integration site sequences constituting ≥1% of all sequences captured are indicated by boxes in ascending order of frequency. Coloured boxes indicate clones identified across time points. Total number of clones identified is listed atop each bar.

the current minimum cGMP requirement of a single biosafety cabinet, centrifuge, incubator and two-person team for each patient product manufactured, a single device requires a fraction of the space and minimal oversight, permitting multiple instruments to process simultaneously. Currently, the semi-automated process described here required a total of 4 h of direct hands-on interaction in smaller time increments separated by each program. Theoretically this could permit technical staff to

**Table 2 | Equivalent transduction efficiency and engraftment of autologous LV gene-modified CD34+ haematopoietic cells manufactured in the semi-automated closed system in myeloablated nonhuman primates.**

| Cell product manufacturing method | Animal ID | Animal weight (kg) | Cell dose infused (cells per kg) | CFCs infused* (dose per kg) | Gene-modified CFCs infused† (dose per kg) | GFP+ cells after 11 days in culture | Days to ANC >500 ml$^{-1}$ | Days to platelets >20,000 ml$^{-1}$ | GFP+ grans in PB at 1 year after transplant | GFP+ lymphs in PB at 1 year after transplant |
|---|---|---|---|---|---|---|---|---|---|---|
| Semi-automated, closed system | Z13105 | 2.7 | $30 \times 10^6$ | $8.0 \times 10^5$ | $2.1 \times 10^5$ | 27.6% | 8 | 23 | 13.2% | 4.5% |
| | Z13083 | 3.7 | $5.1 \times 10^6$ | $4.1 \times 10^4$ | $1.5 \times 10^4$ | 11.9% | 12 | 14 | 25.4% | 14.2% |
| Standard, manual transduction process | J02370 | 4.6 | $11.7 \times 10^6$ | ND | ND | 29.7% | 10 | 35 | 23.3% | 22.1% |
| | A09169 | 4.0 | $11.3 \times 10^6$ | $1.9 \times 10^5$ | $2.1 \times 10^4$ | 8.2% | 7 | 18 | 9.8% | 9.7% |

ANC: absolute neutrophil count; CFC: colony-forming cells; ND: not determined; LV, lentivirus vector; PB: peripheral blood.
*Number of cells in the infusion product with potential to generate colonies in a standard CFC assay. Value is extrapolated from the percentage of colonies generated from 3,000 cells plated in a single assay.
†Number of gene-modified cells in the infusion product with potential to generate colonies in a standard CFC assay. Value is extrapolated from the percentage of gene-modified colonies generated from 3,000 cells seeded in a single assay.

'walk away' from product manufacture intermittently, however, appropriate surveillance systems should be in place to remotely monitor production before this is attempted. Moreover, the closed system components in contact with each cellular product are sterile and disposable, ameliorating the need to extensively decontaminate between patient processes. Automation reduces human error and skill level necessary to manufacture gene-modified cell therapeutics. The ability to administer standardized gene-modified cell therapeutics to more patients will permit more rapid and robust clinical evaluation for commercialization. Moreover, the same programs and protocols can be used to isolate and manufacture other gene-modified blood cell products, such as T cells, or cord blood CD34+ cells (ref. 24 and reviewed in ref. 25), and we have begun testing nuclease-mediated genetic modification of blood cells in this system.

Nearly all gene therapy clinical trials include conditioning to achieve therapeutic levels of gene-modified cell engraftment, requiring a minimum cell dose of $2 \times 10^6$ autologous CD34+ cells per kg of patient body weight at the time of transplant. While extrapolated cell doses from human HPC-A products manufactured in this study would easily achieve this threshold, CD34+ cell numbers from BM products would be more clinically useful for children, requiring further optimization for adult patient populations for whom BM is the desired CD34+ cell source (for example, sickle cell disease). In addition, we observed lower levels of transduction and engraftment of gene-modified cells in the xenotrasplantation model when unmanipulated BM was the source of CD34+ cells, suggesting that culture and transduction conditions will need to be modified to improve efficiency. Alternatively, depletion of mature cells using lineage markers may result in more robust BM cell sources for genetic modification. Thus, we have begun preliminary assessment of a process to deplete lineage marked cells from BM products in this system.

For widespread clinical implementation, reproducible production meeting the safety, quality and sterility specifications ensured by cGMP principles must be shown. We observed a wide range of VCN per cell in each LV gene-modified product, which was observed to be lower in xenotransplanted mice in vivo. This could be due to toxicity induced by high MGMT expression in cells with multiple integrations, as has been previously suggested by Milsom et al.[26]. It may also reflect a reduced engraftment capacity of human cells in a mouse xenotransplant model. Additionally, we determined in vitro the ability to lower VCN per cell when

UM729 is removed from transduction cultures. The molecular mechanism driving high VCN in the presence of UM729 is not understood at this time. Therefore it should be omitted from short-term transduction cultures, but could be investigated more thoroughly for expansion of transduced BM CD34+ cells as a means to increase cell doses available for gene therapy.

In addition, the device culture chamber was not pre-coated with recombinant human fibronectin fragment (RetroNectin), which shows limited efficiency in promoting infectivity of VSV-G pseudotyped LV[27,28], to facilitate cell harvest. This would likely not be possible for other LV vector pseudotypes such as RD114 and gibbon ape leukemia virus, which are dependent on RetroNectin for transduction efficiency. Further refinements to media components, culture and gene modification reagents and conditions are easily implemented in this manufacturing process and could be systematically evaluated with larger numbers of donors to establish statistical significance.

Importantly, this process does not eliminate cGMP manufacture of LV vectors for use in current gene therapy trials. Obtaining consistently high-titre LV is necessary for broad application of gene therapy and stable LV producing cell-line development will be important (reviewed in ref. 29). Also, we have reduced but not eliminated the need for trained staff, as device interfacing during RBC depletion and media exchanges requires subjective input. An optimal system will require the device, a single program with minimal requirement for user interface and a single-use kit consisting of the tubing set and all required components for manufacture of autologous patient cells with minimal staff. Finally, regulatory classification of the medicinal product, autologous gene-modified blood cells, currently differs between USA and the European Union and will need to be addressed.

Despite these challenges, we believe the scalability and utility of this approach provides a major cost advantage. Beyond clinical cost for collecting initial patient products, CD34+ cell-selection reagents and recombinant human growth factors are the most expensive components required, regardless of manufacturing infrastructure or method of genetic modification. Still, a benefit-to-cost ratio should be considered for each gene therapy candidate disease. For example, in HIV+ patients, whose lifetime cost of care is estimated to be US$600,000 (ref. 30), a cell-based curative therapy requiring only a single administration could greatly reduce cost of treatment.

In summary, this study establishes small footprint, semi-automated, mostly closed manufacturing of LV gene-modified CD34$^+$ haematopoietic cells for therapeutic use, conceptualizing a 'point-of-care' delivery approach for gene-modified blood cells using clinically relevant gene transfer in two preclinical transplantation models. This represents a significant advance towards global distribution of HSC gene therapy.

## Methods

**Study design.** The purpose of this study was to demonstrate proof-of-concept that manufacturing of gene-modified CD34$^+$ haematopoietic cells could be performed in a semi-automated, benchtop platform suitable for use at the point of patient care. At least two biological replicates are included for each product-type tested and each was tested and described individually. In addition to two different human CD34$^+$ cell sources, a rigorous evaluation of isolation, gene transfer, engraftment potential and haematopoietic fitness was conducted in NHPs.

**Approved protocols and subjects.** All studies were conducted under protocols approved by the Fred Hutchinson Cancer Research Center and University of Washington Institutional Animal Care and Use Committees and the Fred Hutchinson Cancer Research Center Institutional Review Board in accordance with the Declaration of Helsinki. Cell products were purchased from a commercial source (BM products; HemaCare Corporation or StemExpress) or obtained through institutional shared resources (HPC-A products).

**LV vectors.** The vector used in NHP transplantation (pRSC-SFFV.P140K.PGK.eGFP-sW) is a SIN LV vector produced with a third-generation split packaging system and pseudotyped by the vesicular stomatitis virus G protein (VSV.G). Vector for these studies was produced by our institutional Vector Production Core (PI H.-P.K.). Infectious titre was determined by flow cytometry evaluating eGFP protein expression following titrated transduction of HT1080 human fibrosarcoma-derived cells with research-grade LV vector preparations. The clinical grade anti-HIV vector used in gene transfer to human cell products (pRSC-H1.shCCR5.UbiC.C46.sEf1a.P140K-sW) is also a SIN LV vector pseudotyped with VSV.G protein. Clinical grade vector was produced by the Indiana University Vector Production Facility (IUVPF; Indiana, USA) using a large-scale validated process. The vectors were produced following GMP guidelines under an approved Drug Master File held by IUVPF. Briefly, the anti-HIV LV was produced by means of transient four-plasmid transfection of 293T cells. Unconcentrated vector supernatant was concentrated 200-fold by tangential flow-based purification. A complete description of vector characterization is included in Supplementary Table 7. Infectious titre was measured through transduction of HT1080 cells with serial dilutions of vector and calculation of the copies of integrated vector per cell by quantitative (Taqman) PCR.

**Transplantation procedures.** Pigtailed macaques (*Macaca nemestrina*) were housed at the Washington National Primate Research Center under conditions approved by the American Association for Accreditation of Laboratory Animal Care. Juvenile macaques were primed with G-CSF (100 µg kg$^{-1}$) and stem cell factor (SCF) (50 µg kg$^{-1}$) by subcutaneous injection daily for 5 days, before BM harvest. BM was collected in acid citrate dextrose and heparin under general anaesthesia from both humeri and femora. Autologous CD34$^+$ cells were isolated using biotinylated mouse anti-human CD34 (12.8) antibody produced in the Biologics shared resource of the Fred Hutchinson Cancer Research Center and rat anti-mouse IgM-conjugated microbeads (Miltenyi Biotec) and transduced with the pRSC-SFFV.P140K.PGK.eGFP-sW LV vector at a final MOI of 40 (20 × 2) IU per cell. Following a fractionated dose of 1,020 cGy TBI on day 2 and day 1, autologous gene-modified CD34$^+$ cells were infused back into the animals. Twenty-four hours following infusion of gene-modified cells, animals received intravenous G-CSF (100 µg kg$^{-1}$ day$^{-1}$) until stable neutrophil engraftment (ANC > 0.5 × 10$^9$ l$^{-1}$ (500 µl$^{-1}$)) was attained. Standard supportive care including blood product transfusions, fluid and electrolyte management and antibiotics were administered as needed. Haematopoietic recovery was monitored by daily blood counts. Animals also received oral tacrolimus beginning at 3 mg kg$^{-1}$ day$^{-1}$ to achieve 10–15 ng µl$^{-1}$ in serum as an immunosuppressant to minimize rejection of eGFP-expressing cells. Tacrolimus taper was initiated within 6 months to 1 year after transplant. Samples for *in vivo* gene marking and lineage assessment in the NHP pigtailed macaques were stained at a 1:20 dilution with CD3-PE (552127, clone SP34-2), CD8-Pacific blue (558207, clone RPA-T8), CD20-APC (560836, clone L200), CD13-PE (347837, clone L138) and CD14-APC (555399, M5E2). BM was also stained with anti-human CD34-APC (561209, clone 563). All antibodies were from BD Biosciences.

NOD.Cg-Prkdc scid Il2rγ tm1Wj/Szj (NOD/SCID/IL2rγ$^{null}$, NSG) mice were housed at the Fred Hutchinson Cancer Research Center under conditions approved by the American Association for Accreditation of Laboratory Animal Care. Mice 8–12 weeks old received 270 cGy TBI. Four hours after TBI, mice received 1 × 10$^6$ gene-modified CD34$^+$ cells resuspended in PBS containing 1% heparin (APP

Pharmaceuticals, Schaumburg, IL) via tail vein infusion. Blood samples were collected by retro-orbital puncture over 12 weeks post infusion. At 12 weeks mice were killed and spleen and BM were harvested. Organ samples were filtered through a 70 mm filter (BD Biosciences) and washed with DPBS. Blood and tissue samples were stained with appropriate fluorescent-activated cell sorting (FACS) antibodies for 15 min at room temperature. RBCs were removed by BD FACS Lysing Solution (BD Bioscience). Stained cells were acquired on a FACS Canto II (BD Bioscience) and analysed using FlowJo software v10.0.8 (Tree Star Inc., Ashland, OR). Analysis was performed on up to 20,000 cells in the viable cell population, and gates were established using full minus one stained controls. Samples for *in vivo* gene marking and lineage assessment in NSG mice were stained at a 1:20 dilution using anti-mouse CD45-V500 (561487, clone 30-F11), anti-human CD45-PerCP (347464, clone 2D1), CD3-FITC (555332, clone UCHT1) CD4-V450 (560345, clone RPA-T4), CD8-APCCy7 (557834, clone SK1), CD20-PE (555623, clone 2H7) and CD14-APC (555399, clone M5E2). BM was also stained with anti-human CD34-APC (555824, clone 581). All antibodies were from BD Biosciences.

**Semi-automated mostly closed processing.** For all products, initial blood cell counts and differential analyses were obtained using either a KX-21N (Sysmex) or AcTdiff2 (Coulter) automated haematology analyser. For BM products, initial hematocrit (HCT) > 25% was manually diluted with Plasmalyte A (Baxter). A total of nine programs were developed for semi-automated processing (numbered 1–9), which are described in detail in Supplementary Table 2. Programs are available upon request. For all products, the initial tubing set used (TS100; Miltenyi Biotec) was pre-installed onto the device as part of the initial program setup (either Program 1 or Program 5). Before installation, diluted product, PBS/EDTA buffer (Miltenyi Biotec), HES (Hospira) and diluted cell product were sterile-docked onto the pre-fabricated tubing set. For CD34$^+$ cell enrichment from human products, CliniMACS CD34 reagent (Miltenyi Biotec) was used with intravenous immunoglobulin (10% IVIg; Baxter) as a blocking agent.

For monkey cell products produced using the semi-automated closed system, diluted products were first RBC depleted via HES sedimentation. A two-step labelling protocol was then performed since the anti-CD34 antibody used (12.8) is not directly conjugated to a magnetic bead. Cells were first incubated with anti-CD34 (12.8) antibody and then anti-IgM-conjugated beads. Each labelling step included 30 min incubation at 4 °C with slow rotation in the device chamber. Magnetic column-based selections were performed and resulting products were divided into a Negative Fraction Bag and a Target Cell Bag included in the pre-fabricated tubing set. The TS100 tubing set was then replaced with the TS730 tubing set (Miltenyi Biotec). Complete transduction media consisted of StemSpan animal component-free media (Stem Cell Technologies) containing 100 ng ml$^{-1}$ each recombinant human growth factors SCF (Miltenyi Biotec), thrombopoietin (TPO; PeproTech) and feline sarcoma virus (FMS)-like tyrosine kinase 3 ligand (FLt3L; Miltenyi Biotec), 4 µg ml$^{-1}$ protamine sulfate and 500 nM UM729 (kindly provided by Dr Guy Sauvageau; Université de Montréal). Complete transduction media and concentrated LV were pre-loaded into separate positions on the tubing set via sterile-docking. NHP cells were not prestimulated to divide before transduction. Immediately following exchange into complete transduction media, a pre-determined LV vector volume was added to the cell suspension to begin transduction at a MOI of 20 IU per cell and cells were cultured in the device chamber under 5% CO$_2$ and 37 °C with a gentle mix of the cell suspension every 30 min for an overnight incubation. Twelve hours later, a second, equivalent vector dose was added for a final MOI of 40 IU per cell, as well as additional media to maintain a cell suspension of ∼1 × 10$^6$ cells per ml to continue culture for an additional 4 h. Cells were not rested in between vector additions. For NHP cells, the infectious titre of vector used was 4.1 × 10$^8$ IU ml$^{-1}$ for both animals. Following transduction, the cell suspension was washed to remove media and residual LV vector. Final formulation of the product for infusion included Plasmalyte A containing 5% autologous serum in a 200 ml transfer pack pre-labelled with the animal ID. This product bag was sterile-welded on the device to remove it for final product sampling and transport to infusion.

For monkey cell products produced using standard, operator-dependent methods, diluted products were first RBC depleted via haemolysis in ammonium chloride buffer. Two-step labelling with anti-CD34 (12.8) antibody and then anti-IgM-conjugated beads was manually performed. Each labelling step included 30 min incubation at 4 °C with slow rotation. Magnetic column-based selections were manually performed and resulting products were plated in culture media containing growth factors to prestimulate cell division on tissue-culture treated flasks overnight. The following morning, cells were harvested and plated on flasks pre-coated with retronectin (Takara) at 2 µg cm$^{-2}$ for transduction. Transduction media consisted of either Dulbecco's Modified Eagle's Medium (Gibco; animal J02370) or StemSpan serum-free media (Stem Cell Technologies; animal A09169) containing 4 µg ml$^{-1}$ protamine sulfate and 50–100 ng ml$^{-1}$ each recombinant human growth factors G-CSF, SCF, TPO and FLt3L or a cocktail including interleukin 3 (IL-3; Peprotech), interleukin 6 (IL-6; Peprotech), G-CSF, SCF, TPO and FLt3L. The infectious titres of the LV preparations used in these experiments were 5.1 × 10$^7$ IU ml$^{-1}$ (J02370) and 2.1 × 10$^8$ IU ml$^{-1}$ (A09169). Immediately following exchange into complete transduction media and plating on retronectin-coated flasks, a pre-determined LV vector volume was added to the cell suspension

to begin transduction at a MOI of 10 IU per cell and cells were cultured under 5% $CO_2$ and 37 °C. Six to eight hours later, a second, equivalent vector dose was added for a final MOI of 20 IU per cell and suspensions were incubated overnight. Cells were not rested in between vector additions. The following morning, cell suspensions were washed to remove media and residual LV vector. Final formulation of the product for infusion included Plasmalyte A containing 5% autologous serum.

For human BM products, semi-automated processing was identical to that performed for monkey BM products with the exception that single-step labelling of CD34$^+$ cells was accomplished with a directly conjugated anti-human CD34 magnetic bead available from Miltenyi Biotec.

For human mobilized HPC-A products, no RBC depletion was required. Thus, fewer programs were needed for complete processing. Semi-automated processing included an initial program for preparation, labelling and CD34$^+$ cell enrichment. Following enrichment, transduction culture was performed as described for both monkey and human BM products.

Human cells were not prestimulated to divide in culture before transduction. Instead, following CD34-enrichment, cells were immediately transduced at a MOI of 20 IU per cell as two vector additions of 10 IU per cell ~12 h apart. Cells were not rested in between vector exposures. Complete vector information is included in Supplementary Table 8. Following transduction, cells were washed and formulated for infusion in Plasmalyte A containing 5% human serum albumin (HSA; Baxter) in a 200 ml transfer pack pre-labelled with the autologous subject ID. Complete transduction media for human cells consisted of StemSpan animal component-free media containing 100 ng ml$^{-1}$ each recombinant human growth factors SCF, TPO and FLt3L, 4 µg ml$^{-1}$ protamine sulfate and 500 nM UM729.

**Colony assays and measurement of transduction efficiency.** CFC assays were performed in methylcellulose (H4230; Stem Cell Technologies) containing recombinant human growth factors according to the manufacturer's specifications[19]. Colonies were counted to determine the number of CFC for every 100,000 cells plated. At least 80 individual colonies were picked for each experiment by manual pipetting into sterile tubes containing molecular grade water (HyClone) and protease K (Sigma). Genomic DNA was isolated by incubating tubes at 95 °C for 2 h on a thermal cycler. Crude DNA preparations were then subjected to PCR using LV-specific primers (Fwd: 5′-AGAGATGGGTGCGAGAGCGTCA-3′ and Rev: 5′-TGCCTTGGTGGGTGCTACTCCTAA-3′) and, in a separate reaction, actin-specific primers which were designed for each species (monkey Fwd: 5′-TCC TGTGGCACTCACGAAACT-3′ and Rev: 5′-GAAGCATTTGCGGTGGACGA T-3′ and human Fwd: 5′-TCC TGT GGC ATC GAC GAA ACT-3′ and Rev: 5′-GAAGCATTTGCGGTGGACGAT-3′). Colonies containing expected bands for both LV and actin were scored as transduced. Reactions which did not yield actin products were considered non-evaluable.

**Liquid culture assays and gene marking analysis.** Transduced cells were sub-cultured in Iscove's Modified Dulbecco's Medium containing 10% fetal bovine serum and 1% penicillin/streptomycin and recombinant human growth factors for up to 12 days following manufacture[19]. For analysis of gene marking, leukocytes were isolated from sub-cultures directly, or by ammonium chloride lysis from heparinized PB or BM. For monkey cells transduced with the eGFP-expressing SIN LV, gene marking levels were determined by flow cytometry on a Canto or LSRII cell analysis machine (both from Beckton Dickinson). Flow cytometric data were analysed by FlowJo version 10.0.7 (Tree Star Inc.) or CELLQuest Pro version 5.1 software (Beckton Dickinson). Transgenic eGFP protein expression was determined by gating to exclude fewer than 0.1% of control cells in the relevant region based on forward and right-angle (side) light scatter characteristics or two-colour flow cytometry when cells were stained with antibodies to human CD3 (clone UCHT1), CD4 (clone RPA-T4), CD8 (clone), CD20 (clone 2H7) or CD34 (clone 563). All antibodies were from Beckton Dickinson. For human cells transduced with the clinical anti-HIV LV vector, gene marking was analysed by TaqMan 5′ nuclease quantitative real-time PCR assay. Genomic DNA was extracted using either the Blood DNA Mini kit or the Gentra Puregene Blood kit (both from Qiagen) according to the manufacturer's instructions. Sample DNA was analysed in at least duplicate with a LV-specific primer/probe combination (Fwd: 5′-TGA AAG CGA AAG GGA AAC CA-3′, Rev: 5′-CCG TGC GCG CTT CAG-3, Probe: 5′-AGC TCT CTC GAC GCA GGA CTC GGC-3′). In a separate reaction, a β-globin-specific primer/probe combination was used to adjust for equal loading of genomic DNA per reaction (Fwd: 5′-CCTATCAGAAAGTGGTGGCTGG-3′, Rev: 5′-TTGGACAGCAAGAAAGTGAGCTT-3′, Probe: 5′-TGGCTAATGCCCTGG CCCACAAGTA-3′). Reactions contained genomic DNA, appropriate primer/probe combination, ABI Master Mix (Applied Biosystems) and were run on the ABI Prism 7500 Sequence Detection System (Applied Biosystems) under the following thermal cycling conditions: 50 °C for 2 min and 95 °C for 10 min, then 40 cycles of 95 °C for 15 s and 60 °C for 1 min.

**LV insertion site analysis.** Genomic DNA isolated from bulk leukocytes was subjected to LV-specific amplification of provirus-genome junctions by modified genomic sequencing-PCR[31]. The vector-genome junctions were processed and mapped using either the Vector Integration Site Analysis Server (https://visa.pharmacy.wsu.edu/bioinformatics) for the Genome Reference Consortium build Grhg38 of the human genome[30], or custom PERL scripts for the Beijing Genomics Institute build rheMac3 of the rhesus genome[32]. Sequences that could not be confidently localized to the appropriate genome were removed from the data set before analysis. Clonality was assessed by ranking each unique insertion site by sequence abundance and normalizing to the total non-unique, localized number of insertion site sequence reads recovered for each sample.

**Statistical analysis.** Sample sizes for each product type ($n = 2$ or 3 biological replicates) are too small for meaningful statistical inference.

**Data availability.** The data that support the findings of this study are available from the corresponding author upon reasonable request.

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

## Acknowledgements

We thank the Cellular Processing Facility of Fred Hutch and the Cell Therapy Laboratory of the Seattle Cancer Care Alliance for assistance in planning and conduct of these experiments. We also thank A. Lee, C. Ironside, D. Chandrasekaran, V. Nelson, E. Wilson and K. Sze for assistance in the conduct of these experiments and H. Crawford and B. Larson for help in preparation of the manuscript. We also thank the Indiana University Vector Production Facility for manufacture of clinical grade LV vector. Grant funding for this study included the following: AI097100 (G.D.T.), AI102672 (G.D.T), P30 DK056465 (S.H. and H.-P.K.), P30 CA015704 (S.H.), U19 AI096111 (H.-P.K.), R01 HL116217 (H.-P.K.) and R01 AI080326 (H.-P.K.). H.-P.K. received research funding from the Heath Foundation. H.-P.K. is also a Markey Molecular Medicine Investigator and received support as the inaugural recipient of the Jose Carreras/E. Donnall Thomas Endowed Chair for Cancer Research. J.E.A. also received funding from Fred Hutch.

## Author contributions

J.E.A., T.W., K.G.H., S.P.K., S.H. and H.-P.K. participated in the study design. J.E.A. and T.W designed and implemented custom device programming. J.E.A. and S.P.K. carried out automated manufacturing. J.E.A. and K.G.H. performed all mouse transplantation experiments. J.E.A. performed NHP transplant experiments and follow-up. J.E.A., K.G.H., G.D.T. and J.D.H. participated in the design and execution of clone tracking analyses in mouse and NHP transplant experiments. G.D.T. and J.D.H. provided bioinformatics resources for analysis of LV integration sites analysed in these studies. H.-P.K. manufactured eGFP-expressing LV vector used in these studies and provided clinical grade anti-HIV vector for use in these studies. S.H. provided the mobilized HPC-A products used in these studies. J.E.A. wrote the manuscript. All authors reviewed some or all of the primary data presented in the manuscript and participated in manuscript editing and review.

## Additional information

**Competing financial interests:** T.W. is employed by Miltenyi Biotec. J.E.A has received honoraria from Miltenyi Biotec. The other authors declare no conflicts of interest relevant to this work.

**How to cite this article**: Adair, J. E. *et al.* Semi-automated closed system manufacturing of lentivirus gene-modified haematopoietic stem cells for gene therapy. *Nat. Commun.* **7,** 13173 doi: 10.1038/ncomms13173 (2016).

