## [Peer Review File · Nature Communications]

Reviewer #1 (Remarks to the Author)

This is a very interesting and novel study that described an automated closed system approach for processing bone marrow and mobilized peripheral blood harvests for gene therapy studies. The current means for processing and transducing stem cell grafts is very complicated, time consuming, and requires highly specific technical expertise available only at a limited number of centers throughout the world. Adair and her colleagues described a close automated system for transducing human CD34+ cells that could greatly advance gene therapy by providing consistent, less expensive, and highly reproducible stem cell processing and lentiviral transduction of grafts. It is hard to see how the gene therapy field will advance without an automated process, and this paper provides the first description of a system to accomplish this task. This has broad relevance to the field of stem cell gene therapy, which is now rapidly expanding in academic and industry setting, so the potential significance of this work is considerable.

The system described in this study is based on a semiautomated process that utilizes the CliniMACS Pr4odigy device, which is commercially available. The authors developed a set of custom programming for this device that allows for grafts to undergo 1) RBC depletion, 2) CD34 selection, 3) transduction of selected cells with a lentiviral vector, and 4) harvest and formulation of cells for patient infusion. This can be accomplished with a modest amount of operator input and in limited space. The system was tested with human bone marrow and G-CSF mobilized peripheral blood samples as well as non-human primate bone marrow samples and high quality data is provided regarding the CD34+ cell yields, purity, RBC depletion efficiency, and lentiviral gene marking in transplanted NSG mice (human cells) and transplanted pigtail macaques (NHP samples).

While there is great enthusiasm for the significance and novelty of this study, the effectiveness of this new approach is not clearly established based on the data presented. In general, engraftment and transduction data are variable and at times raise questions regarding the engraftment capacity of the transduced grafts as well as the transduction efficiency. What is lacking in these experiments are controls showing grafts transduced in the conventional manner. Therefore, one cannot be sure whether this automated approach is as efficient as the "gold standard" approach of operator-dependent CD34+ purification and transduction in retroectin coated flasks. A direct comparison of the new technology with the old methodology, done in parallel using the same lentiviral vector and CD34+ cell source, would greatly improve the interpretability of these data as regards the efficiency of the new system. For instance, consider the following points:

- 1) In figure 1b and c, a significant portion of the transplanted NSG mice have very little engraftment with human cells (less than 10%, particularly in APH donor 1). Would this be the same if the cells were processed manually and transduced in flasks ?
- 2) The transduction efficiencies are also relatively low as shown in figure 1d, with most mice showing less than 20% transduction of human cells. Is this due to the automated procedure, or rather intrinsic to the donor source and vector used. A "gold standard", concurrent control is needed to address this question.
- 3) The same criticisms apply to figure 3, very low VCNs in the bone marrow are present in 3C, ... process, cells, or vector ?
- 4) For the monkey experiments shown in figure 4, marking data is only shown for T and B lymphocytes, why is there no data from the myeloid cell populations ? Without myeloid marking data, it is very difficult to interpret the HSC transduction efficiency because lymphocytes are very long-lived in circulation. The data shown in supplementary figure S4 shows very low levels of marking in erythrocytes and platelets, further raising the concern that HSC transduction in these cases was relatively low.

Therefore, it seems critical to provide more experiments to determine if the semi-automated method is achieving similar transduction frequencies to what can be obtained with direct, manual manipulations.

Other points:

- 1) The Title: "Point of Care ..." is very vague and confusing and a scientific reader would have no idea what this means. I recommend changing the title with something that describes the new approach more clearly. Perhaps, "Semi-automated processing and transduction of human HSCs with lentiviral vectors in a close system device" or something along those lines.
- 2) Are the computer programs used in this study going to be made available to the readers ? This is an important question regarding reproducibility of these results in other centers and the ability to advance the field.
- 3) There is not enough description of the transduction process. How many vector applications were used, at what MOI, were the cells washed between transduction, what was the titer of the vector, etc.
- 4) While the clinical anti-HIV P140K vector used is appropriate as a clinical product, would the studies be easier to interpret if a high titer, simple, GFP LV were used ? This would allow better comparison with published transduction efficiencies using standard techniques.

Reviewer #2 (Remarks to the Author)

1. General comment: for RESULTS, it would help the reader if the Bone Marrow Processing section and the APH Processing section were marked as separate in the text.
2. Page 3: why the use of "APH" for a mobilized leukapheresis product? ISBT128 term is HPC-A.
3. Table S1: For samples 3 and 4, please explain the increased yield of CD34+ cells number after RBC depletion? Is it possible that the RBCs inhibited CD34 staining in the pre-depleted samples?
4. Page 5: line 19. Unless you mean this literally, the use of the word "release" is ambiguous and should be "completion".
5. Page 7: line 18-20. It does not seem possible that the cell viability decreased during transduction, despite an increase in viable cell number. Please correct or explain.
6. Page 8, line 15. How was the process modified when the NHP cell product was made?
7. Page 11, line 23. Please provide a reference that supports the lack of effect of fibronectin on VSV-G LV transduction.
8. DISCUSSION. Please comment further on the time of the procedure (~25 hr) and whether the ~4 hr direct involvement of the technician really means that there is 21 hr of 'walk away' time.

POINT-BY-POINT RESPONSE TO REVIEWERS:

Reviewers' comments:

Reviewer #1 (Remarks to the Author):

This is a very interesting and novel study that described an automated closed system approach for processing bone marrow and mobilized peripheral blood harvests for gene therapy studies. The current means for processing and transducing stem cell grafts is very complicated, time consuming, and requires highly specific technical expertise available only at a limited number of centers throughout the world. Adair and her colleagues described a close automated system for transducing human CD34+ cells that could greatly advance gene therapy by providing consistent, less expensive, and highly reproducible stem cell processing and lentiviral transduction of grafts. It is hard to see how the gene therapy field will advance without an automated process, and this paper provides the first description of a system to accomplish this task. This has broad relevance to the field of stem cell gene therapy, which is now rapidly expanding in academic and industry setting, so the potential significance of this work is considerable.

The system described in this study is based on a semiautomated process that utilizes the CliniMACS Prodigy device, which is commercially available. The authors developed a set of custom programming for this device that allows for grafts to undergo 1) RBC depletion, 2) CD34 selection, 3) transduction of selected cells with a lentiviral vector, and 4) harvest and formulation of cells for patient infusion. This can be accomplished with a modest amount of operator input and in limited space. The system was tested with human bone marrow and G-CSF mobilized peripheral blood samples as well as non-human primate bone marrow samples and high quality data is provided regarding the CD34+ cell yields, purity, RBC depletion efficiency, and lentiviral gene marking in transplanted NSG mice (human cells) and transplanted pigtail macaques (NHP samples).

While there is great enthusiasm for the significance and novelty of this study, the effectiveness of this new approach is not clearly established based on the data presented. In general, engraftment and transduction data are variable and at times raise questions regarding the engraftment capacity of the transduced grafts as well as the transduction efficiency. What is lacking in these experiments are controls showing grafts transduced in the conventional manner. Therefore, one cannot be sure whether this automated approach is as efficient as the "gold standard" approach of operator-dependent CD34+ purification and transduction in retroectin coated flasks. A direct comparison of the new technology with the old methodology, done in parallel using the same lentiviral vector and CD34+ cell source, would greatly improve the interpretability of these data as regards the efficiency of the new system.

RESPONSE: The intention of our study was not to demonstrate the inferiority of manual processing of autologous grafts for gene therapy approaches, but to establish the proof of concept that this can be done in an automated setting, which, as this reviewer contends, will be necessary for the field of gene therapy to advance. We believe that the effectiveness of this approach is clearly established based on the data we present demonstrating manufacture transduced CD34⁺ cell populations which not only engraft in autologous nonhuman primate recipients in the myeloablative setting, but at levels equivalent to previous reports with VSV-g pseudotyped lentivirus vectors transduced using traditional methods including manual processing and transduction on retroectin at the same multiplicity of infection. To emphasize this in the revised manuscript, we have added experimental data comparing standard transduction of CD34⁺ cells to our automated approach in the nonhuman primate. We compared the two animals presented in the original manuscript to two separate animals that received CD34⁺ cells transduced with the exact same LV vector under standard, manual conditions in our laboratory. We observed equivalent levels of transduction, hematopoietic reconstitution and gene modified cell engraftment across these four animals. We have summarized this comparison into the table shown below, which is now included in the main body of the revised manuscript as **Table 2**. Per this reviewer's request, this replaces the previous Supplemental Table S7 since it represents a critical control for these studies.

Table 2. Equivalent transduction efficiency and engraftment of autologous, LV gene modified CD34⁺ hematopoietic cells manufactured in the semi-automated, closed system in myeloablated nonhuman primates.

Cell Product Manufacturing Method	Animal ID	Animal Weight (kg)	Cell dose infused (cells/kg)	CFCs infused [†] (dose/kg)	Gene modified CFCs infused ^ε (dose/kg)	GFP ⁺ Cells After 11 Days in Culture	Days to ANC >500/mcL	Days to Platelets >20,000/mcL	GFP ⁺ Grans in PB at 1 year After Transplant	GFP ⁺ Lymphs in PB at 1 year After Transplant
Semi-Automated, Closed System	Z13105	2.7	30 × 10 ⁶	7.98 × 10 ⁵	2.1 × 10 ⁵	27.6%	8	23	13.2%	4.5%
	Z13083	3.7	5.1 × 10 ⁶	4.1 × 10 ⁴	1.5 × 10 ⁴	11.9%	12	14	25.4%	14.2%
Standard, Manual Transduction Process	J02370	4.6	11.7 × 10 ⁶	ND	ND	29.7%	10	35	23.3%	22.1%
	A09169	4.0	11.3 × 10 ⁶	1.9 × 10 ⁵	2.1 × 10 ⁴	8.2%	7	18	9.8%	9.7%

PB: peripheral blood; ND: Not determined; CFC: colony-forming cells; ANC: absolute neutrophil count

[†] Number of cells in the infusion product with potential to generate colonies in a standard CFC assay. Value is extrapolated from the percentage of colonies generated from 3000 cells plated in a single assay.

^ε Number of gene modified cells in the infusion product with potential to generate colonies in a standard CFC assay. Value is extrapolated from the *percentage* of gene modified colonies generated from 3000 cells seeded in a single assay.

Reviewer Comment: For instance, consider the following points:

1) In figure 1b and c, a significant portion of the transplanted NSG mice have very little engraftment with human cells (less than 10%, particularly in APH donor 1). Would this be the same if the cells were processed manually and transduced in flasks?

RESPONSE: To address this reviewer's concern that a significant proportion of the NSG mice show very little engraftment with cells selected and processed in the closed system described here, we have compared our data to NSG mice transplanted by another laboratory in the published literature. We have cited Greene, M., *et al. Human Gene Therapy Methods*, 2012 Part B, Methods. 23: 297-308 (PMID:23075105) wherein the authors transduced human CD34⁺ hematopoietic cells from mobilized apheresis products with a VSV-G pseudotyped LV vector expressing the human gamma globin gene at a multiplicity of infection of 135 infectious units/cell and transplanted these into NSG mice at 1 × 10⁶ cells per mouse following conditioning with 35 mg/kg busulfan. The authors observed median engraftment of 22.5% human CD45⁺ cells in the bone marrow of these mice at 12–16 weeks after transplantation, with a mean vector copy number ranging between 0.15 and 0.25 copies per human cell. In comparison, we observed 21% and 46% human CD45⁺ cells in the bone marrow of mice transplanted in our study (**original Figure 1C**) with VCN in human cells ranging from 0.1 to 0.17 per cell. Importantly, our levels of gene modified cells are only modestly lower to that reported by Greene and colleagues despite a dramatically lower MOI of 20 IU/cell during transduction, nearly 7-fold lower than that used in the published report. For reference, we have excerpted the relevant figures from Greene *et al.* and our current data below for direct comparison.

2) The transduction efficiencies are also relatively low as shown in figure 1d, with most mice showing less than 20% transduction of human cells. Is this due to the automated procedure, or rather intrinsic to the donor source and vector used. A "gold standard", concurrent control is needed to address this question.

RESPONSE: As demonstrated in the above responses, we believe the levels of transduced cells observed in xenotransplanted mice are in line with previously published studies performing similar experiments (Greene *et al.*, 2012) at nearly seven times the multiplicity of infection used in our study. Importantly, we would also like to emphasize the autologous nonhuman primate model data presented in our manuscript as noted in the first response above and in Table 2 of the revised manuscript. In this clinically relevant setting, we did not observe reduced engraftment of transduced autologous cells in the two animals who received cells manufactured in the automated system in this study compared to two separate animals that received CD34⁺ cells transduced with the exact same LV vector under standard, manual conditions in our laboratory. Again, this data is now included in **Table 2**. Per this reviewer's request, this replaces the previous Supplemental Table S7 since it represents a critical control for these studies.

3) The same criticisms apply to figure 3, very low VCNs in the bone marrow are present in 3C, .. process, cells, or vector?

RESPONSE: We acknowledge in the Results and Discussion sections of the original manuscript that processing of CD34⁺ cells from steady state bone marrow products for LV vector transduction was not as efficient as observed for mobilized apheresis products. However, as noted in the above response to points 1 and 2, the human CD45⁺ cell engraftment and VCN in human cells observed in these studies are not dramatically different than what was previously reported by Greene and colleagues after LV transduction of mobilized peripheral blood CD34⁺ cells at seven times the MOI used in the current studies. There is no equivalent data on LV transduced steady state bone marrow derived CD34⁺ cells that we are aware

of in the published literature. Therefore, we do not have a cited reference for comparison here. To address the potential difference between CD34⁺ cell sources, we have revised the text in the discussion section of the manuscript as follows:

While extrapolated cell doses from human HPC-A products manufactured in this study would easily achieve this threshold, CD34⁺ cell numbers from BM products would be more clinically useful for children, requiring further optimization for adult patient populations for whom BM is the desired CD34⁺ cell source (e.g. sickle cell disease). Additionally, we observed lower levels of transduction and engraftment of gene modified cells in the xenotransplantation model when unmanipulated BM was the source of CD34⁺ cells, suggesting that culture and transduction conditions will need to be modified to improve efficiency.

4) *For the monkey experiments shown in figure 4, marking data is only shown for T and B lymphocytes, why is there no data from the myeloid cell populations? Without myeloid marking data, it is very difficult to interpret the HSC transduction efficiency because lymphocytes are very long-lived in circulation. The data shown in supplementary figure S4 shows very low levels of marking in erythrocytes and platelets, further raising the concern that HSC transduction in these cases was relatively low.*

Therefore, it seems critical to provide more experiments to determine if the semi-automated method is achieving similar transduction frequencies to what can be obtained with direct, manual manipulations.

RESPONSE: We would like to clarify that gene marking in both granulocytes (myeloid) and lymphocyte compartments of nonhuman primates is shown in the graphs included in Figures 4a and 4b of the original manuscript (green lines). We have clarified this in the results section of the revised manuscript. Figures 4c and 4d were meant to further classify gene modification levels in lymphocyte sub-populations since T cell engraftment is critical for the target disease referenced in this study, (i.e. HIV/AIDS). To further support the successful transduction and engraftment of HSC, we have now extended these analyses in the revised manuscript to more than 450 days after transplant for each of these animals. Here, HSC transduction and engraftment is supported by long-term, stable levels of gene modification in both granulocytes and lymphocytes at > 1 year since transplant. We have also included engraftment levels of autologous LV gene modified cells transplanted into two control nonhuman primates after standard culture and transduction for comparison as shown in the response to point 2 above (**Table 2** in the revised manuscript). This comparison demonstrates equivalent engraftment of LV gene modified cells using the point-of-care approach compared to standard manufacturing in a highly clinically relevant large animal model of autologous HSC transplantation in the myeloablative setting.

Other points:

1) *The Title: "Point of Care ..." is very vague and confusing and a scientific reader would have no idea what this means. I recommend changing the title with something that describes the new approach more clearly. Perhaps, "Semi-automated processing and transduction of human HSCs with lentiviral vectors in a close system device" or something along those lines.*

RESPONSE: We have revised the title to "Semi-automated, closed system manufacturing of lentivirus gene modified hematopoietic stem cells for gene therapy".

2) *Are the computer programs used in this study going to be made available to the readers ? This is an important question regarding reproducibility of these results in other centers and the ability to advance the field.*

RESPONSE: A statement has been added to the revised manuscript to indicate that programs will be made available upon request.

3) *There is not enough description of the transduction process. How many vector applications were used, at what MOI, were the cells washed between transduction, what was the titer of the vector, etc.*

RESPONSE: We believe that all of the requested components of the transduction process were included in the original manuscript, but were scattered throughout the document. To address this, we have revised the supplemental online materials and methods section of the manuscript to clarify and centrally locate all transduction procedures in a single manuscript section.

4) *While the clinical anti-HIV P140K vector used is appropriate as a clinical product, would the studies be easier to interpret if a high titer, simple, GFP LV were used ? This would allow better comparison with published transduction efficiencies using standard techniques.*

RESPONSE: To clarify, a simplified GFP-encoding LV vector was used in the nonhuman primate studies reported in the original manuscript. For experiments in human CD34⁺ cells, we wanted to demonstrate the same feasibility with a clinical grade LV vector. The titer of the GFP-LV used in these experiments is 1.2×10^{10} IU/mL based on infectious titer observed in HT1080 cells, compared to the clinical vector titer of 7.7×10^8 IU/mL used in the originally reported experiments. Both infectious titers are clinically relevant for current gene therapy trials.

Reviewer #2 (Remarks to the Author):

1. *General comment: for RESULTS, it would help the reader if the Bone Marrow Processing section and the APH Processing section were marked as separate in the text.*

RESPONSE: We have separated these results in the revised manuscript as requested.

2. *Page 3: why the use of "APH" for a mobilized leukapheresis product? ISBT128 term is HPC-A.*

RESPONSE: We have revised the manuscript to utilize the ISBT128 term as requested.

3. *Table S1: For samples 3 and 4, please explain the increased yield of CD34+ cells number after RBC depletion? Is it possible that the RBCs inhibited CD34 staining in the pre-depleted samples?*

RESPONSE: We are not certain of the mechanism by which CD34⁺ cell numbers increase following RBC depletion and can only comment that this observation is not consistent across all donor products. It is possible, as the reviewer suggests, that RBCs may interfere with CD34 staining in the original product sample used for flow cytometry; however, in this study RBC are lysed prior to staining and flow cytometry analyses. Another possible explanation for this observation is potential cell clumping in the original sample, as the total number of white blood cells is also observed to increase in at least one of these samples after RBC depletion (sample 3). In this case, the process of RBC depletion may promote a more homogenous single cell suspension, permitting greater numbers of CD34⁺ cells to be enumerated. Regardless, we do not observe higher CD34⁺ cell numbers following immunomagnetic bead-based enrichment. Thus, further investigation into this phenomenon, we believe, is outside the scope of the current report.

4. *Page 5: line 19. Unless you mean this literally, the use of the word "release" is ambiguous and should be "completion".*

RESPONSE: We have revised the manuscript to change this reference as requested with some additional clarification for readers.

5. Page 7: line 18-20. It does not seem possible that the cell viability decreased during transduction, despite an increase in viable cell number. Please correct or explain.

RESPONSE: Here, we were trying to convey that absolute cell numbers expanded during transduction culture, but the overall viability of this population as measured by trypan blue exclusion was suboptimal. This is important as overall viability of the manufactured product for infusion is often required to be >50% in order for the cell population to be cleared for patient administration. To clarify, we have revised the manuscript with the following statements:

Here, we observed expansion of the cell population during transduction culture, documented as an increase in total cell numbers (from 9.3×10^6 to 30×10^6 cells). However, the overall viability of the resulting cell product after final formulation as measured by trypan blue dye exclusion was 40%, suggesting processing during transduction or harvest and formulation resulted in suboptimal health.

6. Page 8, line 15. How was the process modified when the NHP cell product was made?

RESPONSE: We have revised the supplemental online *Materials and Methods* section of the manuscript to clarify all transduction procedures. The only differences between the human cell product and the nonhuman primate cell product processes were the labeling procedure for CD34⁺ cell enrichment, the vector and MOI used during transduction of nonhuman primate cells. These differences are explained in the original manuscript *Results* section and have been retained in the revised version of the manuscript. Specifically, the human CD34 bead reagent available for clinical separation of CD34⁺ cells in human blood and marrow products does not cross-react with nonhuman primate CD34⁺ cells. Thus, we used a cross-reactive antibody (clone 12.8), that recognizes a different epitope. This antibody is not directly conjugated to the magnetic beads used for separation. Thus labeling of nonhuman primate CD34⁺ cells for immunomagnetic separation requires two stages, one incubation with anti-CD34(12.8) antibody, followed by a second incubation with anti-IgM-conjugated magnetic beads. All other components of the selection process were the same. For transduction of nonhuman primate cells, a green fluorescent protein (GFP)-expressing LV vector was used. This LV vector contains an identical backbone to the clinical anti-HIV vector used in human cell transductions. The MOI used for nonhuman primate cell transductions was 40 IU/cell in total, whereas the MOI used for human cell transductions was 20 IU/cell in total. The reason for the lower MOI in human cells was solely to conserve the clinical grade vector volume available for our ongoing anti-HIV gene therapy clinical trial. Both MOIs used are representative of MOIs applied in current clinical trials of LV gene therapy, albeit even lower than those used by most groups (see Greene, M. *et al. Human Gene Therapy Methods*, 2012 Part B, Methods. 23: 297-308 (PMID:23075105) for MOI = 135 IU/cell; Aiuti A., *et al. Science*, 2013 341: 1233151 (PMID: 23845947) for MOI = 200 IU/cell; Biffi A., *et al. Science*, 2013 341: 1233158 (PMID: 23845948) for MOI = 200 IU/cell).

7. Page 11, line 23. Please provide a reference that supports the lack of effect of fibronectin on VSV-G LV transduction.

RESPONSE: We have included the following two references in the revised manuscript which document the reduced efficiency of recombinant fibronectin fragment in promoting infectivity of VSV-G pseudotyped LV vectors compared to GALVTR-LV and RD114TR-LV and have revised to text to more accurately reflect these observations.

1. Sandrin, V, Boson, B, Salmon, P, Gay, W, Nègre, D, Le Grand, R *et al.* (2002). Lentiviral vectors pseudotyped with a modified RD114 envelope glycoprotein show increased stability in sera and augmented transduction of primary lymphocytes and CD34⁺ cells derived from human and nonhuman primates. *Blood* 100: 823–832.
2. Haas, DL, Case, SS, Crooks, GM and Kohn, DB (2000). Critical factors influencing stable transduction of human CD34⁺ cells with HIV-1-derived lentiviral vectors. *Mol Ther* 2: 71–80.

8. DISCUSSION. Please comment further on the time of the procedure (~25 hr) and whether the ~4 hr direct involvement of the technician really means that there is 21 hr of 'walk away' time.

RESPONSE: We have revised the *Discussion* section of the manuscript to clarify that this could be considered “walk away” time but that it is not a continuous 21 hours. Furthermore, to reliably continue manufacturing during the hands-off time without direct oversight, an alarm and status notification system should be in place to remotely track progress of the semi-automated system.

Reviewer #1 (Remarks to the Author)

The comparison shown in the newly included Table 2 addresses my main concern that the automated process had not been compared to the standard manual transduction procedure. This shows that in transplanted non-human primates, the transduction and engraftment with the automated process is giving equivalent results. The other changes that have been made also result in an improved manuscript including a more direct title, description of the key transduction parameters used in the automated process, and inclusion of extended reconstitution data in the primate system.

POINT-BY-POINT RESPONSE TO REVIEWERS:

Reviewers' comments:

Reviewer #1 (Remarks to the Author):

The comparison shown in the newly included Table 2 addresses my main concern that the automated process had not been compared to the standard manual transduction procedure. This shows that in transplanted non-human primates, the transduction and engraftment with the automated process is giving equivalent results. The other changes that have been made also result in an improved manuscript including a more direct title, description of the key transduction parameters used in the automated process, and inclusion of extended reconstitution data in the primate system.

RESPONSE: The reviewer has acknowledged the changes made to the manuscript, and had no further suggestions. We thank the reviewer for their time and thoughtful review, which have improved this manuscript.